# Neuroprotective Effect of Platinum Nanoparticles Is Not Associated with Their Accumulation in the Brain of Rats

**DOI:** 10.3390/jfb14070348

**Published:** 2023-06-29

**Authors:** Alexander Gennadievich Filippov, Valery Vasil’evich Alexandrin, Alexander Vladimirovich Ivanov, Alexander Alexandrovich Paltsyn, Nadezhda Borisovna Sviridkina, Edward Danielevich Virus, Polina Olegovna Bulgakova, Joanna Petrovna Burmiy, Aslan Amirkhanovich Kubatiev

**Affiliations:** 1Institute of General Pathology and Pathophysiology, Baltiyskaya St., 8, 125315 Moscow, Russia; algf@yandex.ru (A.G.F.); aleksandrin-54@mail.ru (V.V.A.); lrrp@mail.ru (A.A.P.); mag115@list.ru (N.B.S.); edwardvirus@yandex.ru (E.D.V.); polya.bulgakova@gmail.com (P.O.B.); niiopp@mail.ru (A.A.K.); 2Russian Medical Academy for Continuing Professional Education, Barricadnaya St., 2/1 b. 1, 125993 Moscow, Russia; 3Institute of Microelectronic Technology and Ultra-High-Purity Materials, Akademika Osip’yana Str., 6, 142432 Chernogolovka, Russia; burmii@iptm.ru

**Keywords:** aminothiols, cerebral blood flow, cerebral ischemia, cysteine, glutathione, homocysteine, platinum nanoparticles, rat, S-adenosylhomocysteine, S-adenosylmethionine

## Abstract

Platinum nanoparticles (nPts) have neuroprotective/antioxidant properties, but the mechanisms of their action in cerebrovascular disease remain unclear. We investigated the brain bioavailability of nPts and their effects on brain damage, cerebral blood flow (CBF), and development of brain and systemic oxidative stress (OS) in a model of cerebral ischemia (hemorrhage + temporary bilateral common carotid artery occlusion, tBCAO) in rats. The nPts (0.04 g/L, 3 ± 1 nm diameter) were administered to rats (*N* = 19) intraperitoneally at the start of blood reperfusion. Measurement of CBF via laser Doppler flowmetry revealed that the nPts caused a rapid attenuation of postischemic hypoperfusion. The nPts attenuated the apoptosis of hippocampal neurons, the decrease in reduced aminothiols level in plasma, and the glutathione redox status in the brain, which were induced by tBCAO. The content of Pt in the brain was extremely low (≤1 ng/g). Thus, nPts, despite the extremely low brain bioavailability, can attenuate the development of brain OS, CBF dysregulation, and neuronal apoptosis. This may indicate that the neuroprotective effects of nPts are due to indirect mechanisms rather than direct activity in the brain tissue. Research on such mechanisms may offer a promising trend in the treatment of acute disorders of CBF.

## 1. Introduction

Disorders of cerebral circulation are one of the major global causes of death and disability [1]. An impaired cerebral blood flow (CBF) plays an important role in the pathogenesis of secondary brain damage caused by stroke, injury, or cardiac arrest [2,3]. One of the most common and clinically important indicators of CBF disorders is the occurrence of postischemic brain hypoperfusion despite the restoration of blood supply to ischemic sites because of the increased resistance of cerebral vessels. Postischemic hypoperfusion is considered a major factor in the spread of edema during stroke [4]. Therefore, the search for new approaches to reducing secondary brain damage is important.

To date, there are essentially two concepts for the treatment of acute CBF disorders. The first involves the use of thrombolytics (recombinant tissue plasminogen activators) or endovascular thrombectomy, which enables recanalization and restoration of blood flow in damaged vessels. The effectiveness of this approach in practice is significantly limited by the time window available (up to 3–4.5 h for thrombolysis and 6–24 h for endovascular thrombectomy) and the possibility of severe complications or contraindications after use [5,6]. The second concept involves the use of various agents that have a neuroprotective effect.

The brain has a high rate of oxygen consumption, and unlike most organs, it has no internal energy reserves. This causes a rapid development of oxidative stress (OS) in the brain, a process in which the generation of reactive oxygen species (ROS) prevails over the ability to eliminate them under conditions of ischemia and during subsequent reperfusion. Also, cerebral ischemia causes rapid and strong activation of the sympathetic nervous system [7], which apparently triggers the development of systemic endothelial dysfunction (ED) [8]. Due to the key role of OS in the development of ischemic brain damage [9], ED [10], and postischemic hypoperfusion [11], the use of antioxidants is one of the promising areas of neuroprotection. For example, there have already been a few clinical studies showing the effectiveness of the low-molecular-weight antioxidants edaravone [12] and N-acetylcysteine [13].

The development of nanotechnology has led to the emergence of new approaches for the treatment and diagnosis of cerebrovascular diseases based on the use of biocompatible nanoparticles as transporters of drugs or diagnostic probes [14,15]. For instance, there are nanoparticles of noble metals (including platinum (nPts), gold, titanium, and palladium) that have catalytic activity. Unlike the salts of these metals, nanoparticles are not toxic, although the details of their metabolism are not clear [16]. The catalytic activity of nPts has been demonstrated for a wide range of redox reactions, including the decomposition of ROS such as H_2_O_2_, OH^•^, and O_2_^−^ [17,18]. However, nPts can also catalyze the oxidation of biomolecules.

Low-molecular-weight aminothiols (cysteine—Cys, glutathione—GSH, homocysteine—Hcy, and others) are highly sensitive to OS. In response to cerebral ischemia, there is a rapid decrease in the content of their reduced forms in blood plasma as well as a drop in the redox status of the main intracellular antioxidant (GSH) in the nervous tissue itself [19,20]. GSH has previously been shown to be associated with stroke severity [21]. The important protective role of GSH was also confirmed by the efficacy of using N-acetylcysteine (a readily available substrate for GSH synthesis) in models of ischemia [22,23] and in the treatment of stroke [13]. An elevated Hcy level, the damaging effect of which is associated with the induction of OS [24], is considered a risk factor for stroke since a decrease in Hcy level can reduce this risk [25]. The effect of nPts on the homeostasis of aminothiols in cells is still poorly understood. Previously, it was shown that nPts decreased the cellular GSH level, and this effect was correlated with the particle size in an inverse manner but appeared not to be based on the formation of ROS [26]. To the best of our knowledge, studies on the effect of nPts on aminothiols under stress conditions have not yet been carried out.

Under ischemia–reperfusion injury, the development of OS leads to the activation of a number of transmethylases in the nervous tissue, which manifests as hypermethylation of DNA, noncoding RNA, and histones, as well as disruption of the biosynthesis of polyamines and acetylcholine, which in turn enhance inflammatory damage to neurons [27]. During these transmethylation reactions, S-adenosylmethionine (SAM) is consumed and S-adenosylhomocysteine (SAH) is formed. Thus, the SAM/SAH ratio is referred to as the global methylation index [28]. Numerous works involving models of cerebral ischemia have shown a protective effect of SAM administration [29,30,31], including decreasing the GSH content in the brain [32]. However, studies on the effect of nPts on the balance of SAM and SAH have not yet been conducted.

Although the metabolism of nPts and the range of their biological effects remain poorly understood, recent models proposed that the main mechanism by which inflammatory reactions are inhibited in the presence of nPts is the suppression of OS in cells [16]. A previous study showed that the introduction of nPts mitigated damage to brain tissues caused by a 1 h occlusion of the mesencephalic artery in rats. The introduction of nPts led to an improvement in postischemic motor function and a decrease in the size of infarction in the brain cortex. These results indicated that the degradation of collagen IV, the activation of metalloproteinase-9, and the production of O_2_^−^ in the ischemic area were all inhibited [33]. In addition, the introduction of nPts reduced the negative side effects of stroke treatment via tissue plasminogen activator in the same model [34].

In the study on the neuroprotective effect of substances, one of the key issues is their bioavailability; i.e., the ability to penetrate through the blood–brain barrier (BBB), which is a complex of membranes and intercellular contacts formed by the endothelium of blood vessels, astrocytes, and pericytes [35]. The BBB prevents the transfer of many substances from the blood to the brain. Although it has previously been shown that metal (in particular, gold) nanoparticles penetrate the BBB through the mechanisms of passive diffusion, carrier-mediated transport, adsorptive-mediated endocytosis, or pinocytosis [36], there are no data yet on how the BBB can be made permeable to nPts.

Thus, based on the neuroprotective effect of nPts in vivo, we set out to shed light on (1) whether nPts are able to effectively suppress systemic OS and postischemic hypoperfusion induced by acute impairment of cerebral blood flow, and (2) whether the effect of nPts is due to a direct effect on brain tissue or due to the extracerebral effect of nanoparticles. To achieve these objectives, we determined indicators of systemic (reduced plasma levels of aminothiols) and brain (reduced GSH, SAM, and SAH in the hippocampus) OS, CBF, and nPt content using a model of temporary global cerebral ischemia in rats.

## 2. Materials and Methods

### 2.1. Animals and Experimental Design

Male outbred white rats (300–350 g) were used in the experiments. The rats were housed under conditions of controlled temperature (19–25 °C) and humidity (30–70%) in Macrolin cages with food and water available ad libitum. Light was on from 7:00 a.m. until 7:00 p.m. All experimental procedures involving rats were approved by the Ethical Committee at the Institute of General Pathology and Pathophysiology and were carried out in accordance with the recommendations in the Animal Care and Use Committee guidelines.

The animals were randomly divided into four groups. Then, 0.9% NaCl was administered to the first group (*n* = 10) and nPts to the second group (*n* = 10). These rats underwent a sham operation (control). The third (*n* = 19) and fourth (*n* = 10) groups underwent temporary bilateral occlusion of the common carotid arteries (tBCAO) with the administration of nPts or 0.9% NaCl, respectively. Five rats from each group were euthanized for morphological examination of the brain at 3 and 7 days after tBCAO. Five rats from each group were euthanized for determination of reduced low-molecular-weight aminothiols (rCys, rGSH, and rHcy) in blood plasma at 3 h after reperfusion. Five rats from each group were euthanized for determination of reduced GSH and oxidized GSH, SAM, and SAH in the hippocampus at 3 h and 3 days after reperfusion. Nine rats from the third group were euthanized for determination of nPts in the hippocampus, blood, and liver at 3 h after reperfusion. The outline of the experiment is presented in Figure 1.

### 2.2. Surgical Technique (tBCAO) and CBF Determination

The animals were anesthetized with 50 mg/kg of pentobarbital sodium (Nembutal^®^, Ovation Pharmaceuticals, Deerfield, IL, USA). The depth of anesthesia was assessed via an absence of response of the vibrissae to a pain stimulus. The femoral artery was catheterized, and heparin was administered intra-arterially at a dose of 500 U/kg to allow the measurement of the MAP and taking of blood samples from the animals. Rectal temperature was maintained close to 36.7 °C using a heat lamp. The animals were euthanized with an overdose of pentobarbital sodium at the end of the experiments.

The tBCAO was induced as described previously [19]. Systemic blood pressure was reduced by 40–45 mm Hg by inducing hemorrhage (approx. 30% of blood volume or 2.5 ± 0.2 mL/100 g body weight). This was followed by bilateral occlusion of the common carotid arteries for 10 min followed by reinfusion of blood and removal of the carotid artery clips. In the control group, a sham procedure was performed without blood loss or compression of blood vessels. One milliliter of nPts (0.22 mM in 0.9% NaCl) or 0.9% NaCl was administered intraperitoneally at the beginning of reperfusion. CBF was measured using the laser Doppler flowmetry method before the start of tBCAO, during ischemia, and 120 min after reperfusion, as described previously [37]. The LAZMA software (v. 2.2.0.507; LAZMA, Moscow, Russia) was used for CBF analysis.

### 2.3. Synthesis and Characteristics of nPt

H_2_PtCl_6_·6H_2_O, trisodium citrate dihydrate, NaBH_4_, polyvinylpyrrolidone (PVP) with a low molecular weight of 12,600 ± 2700 g/mol, and 0.9% NaCl solution in water for infusion were used for synthesis. A total of 2 mL of sodium citrate (28 mg/mL) aqueous solution, 2 mL of PVP aqueous solution (10 mg/mL), and 2 mL of NaBH_4_ (3 mg/mL) were added to 17 mL of H_2_PtCl_6_ aqueous solution (0.15 mg/mL). The efficiency of Pt reduction was evaluated using the reaction in which a colored complex of Pt ions with iodide ions is formed in acidic conditions [38]. To obtain a formulation suitable for biomedical applications, nPts were removed from the reaction mixture via precipitation and ultracentrifugation at 140,000× *g* for 40 min, resuspended in physiological solution, and then filtered through a membrane with a 0.22 μm pore diameter. For the animal experiments, an nPt solution with a Pt concentration of 0.22 mM (0.04 mg/mL) was used.

Electron micrographs and electron diffraction patterns were obtained via transmission electron microscopy using a LEO 912 AB OMEGA microscope (Carl Zeiss, Jena, Germany). The samples for analysis were prepared by applying a drop of the nPt solution to a copper mesh and drying it in air. A Zetasizer Nano ZS photonic particle analyzer (Malvern, UK) was used to determine the size and ζ potential of particles. The analyzer had a particle measurement range of 0.6 to 6000 nm. The operating temperature range was 2–120 °C, the angle of detection of scattered light was 173 °C, a helium–neon laser with a wavelength of 633 nm was used as a light source, and the power of the light source was 5 MW. The device determined particle sizes by measuring the rate of fluctuation of scattered light by particles. The measurement was carried out in automatic mode according to the standard procedure. The glass cuvette was filled with 1 mL of the sample and loaded into the cuvette compartment of the device. The beam of light emitted by the laser passed through the attenuator and entered the sample cell. The light scattered by particles was detected by the detector. The electrical signal of the detector, which is proportional to the light intensity, was processed by the correlator according to mathematical algorithms embedded in the software. When determining the ζ potential, an immersion-type electrode was lowered into the cuvette filled with the sample. The sample was exposed to an electric field, and the electrophoretic mobility of particles in the electric field was used to calculate the ζ potential. The solvent was water.

### 2.4. Determination of Pt Levels in Tissues

The Pt levels in blood plasma, liver, and brain samples were determined using an X-series inductively coupled plasma mass spectrometer (ICP-MS; Thermo Scientific, Waltham, MA, USA) as described previously [39]. Sample preparation was performed via acid decomposition of the samples in an autoclave with resistive heating [40].

### 2.5. Morphological Examination of the Brain

A morphological examination was performed at 3 and 7 days after tBCAO. In anesthetized rats, the thorax was opened, and perfusion of the brain was performed transcardially through the left ventricle using 20 mL of isotonic sodium chloride followed by 20 mL of 2.5% glutaraldehyde in phosphate-buffered saline. The brain extracted from the skull was additionally fixed and stored in a 2.5% solution of glutaraldehyde at a temperature of +4 °C. Tissues were sampled from the hippocampus via a cut of 2–5 mm from the bregma in the caudal direction [41]. Then, 10 µm thick hippocampal sagittal sections were obtained using a vibratome (Leica VT1000S) and stained with Luxol fast blue Kluver and Barrera (Bio-Optica). The sections were dehydrated in absolute alcohol and covered with coverslips. Images of the dentate gyrus of the hippocampus were taken using an Olympus BX51 microscope. Quantification was performed using the cellSens Standard software Ver. 2.3. It consisted of determining the content of hyperchromic neurons in the sections as a percentage of the total number of neurons in the studied area of the dentate gyrus.

### 2.6. Determination of Reduced Low-Molecular-Weight Aminothiols

Blood samples (1 mL) were obtained from the tail vein 3 h after reperfusion. Venous blood was collected into tubes containing sodium citrate and centrifuged at 4500× *g* for 3 min. The blood plasma was collected and processed as described previously [20], frozen at −80 °C, and stored until analysis.

The brains were removed, and the hippocampus was isolated at 3 h and 3 days after reperfusion. Brain slices (20–50 mg) were prepared and homogenized in acetonitrile with 5,5′-Dithiobis (2-nitrobenzoic) acid (DTNB) or iodoacetamide (IAA) for rGSH and oxidized GSH determination, respectively. A total of 10 mL of extragent (20 mM of DTNB with 2.5 mM of internal standard penicillamine (PA) or 5 mM of IAA) was added per 1 g of tissue. The brain slices (20–50 mg) were homogenized in 10% HClO_4_ (10 mL per 1 g) for SAM/SAH analysis. The probes were centrifuged for 10 min at 15,000× *g*, and the acetonitrile-containing supernatants (100 μL) were dried under vacuum (45 min at 60 °C). Pellets were resuspended in 0.1 M of Na-phosphate buffer with a pH of 7.4 before analysis using ultra-performance liquid chromatography (UPLC) as described previously [20].

We used an UPLC H-class ACQUITY system (Waters, Milford, MA, USA) with a PDAλ UV detector (λ = 330 nm) to measure the concentrations of reduced low-molecular-weight thiols. Ten microliters of each sample was injected onto an Eclipse Plus C18 100 × 2.1 mm × 1.8 μm column (Agilent, Santa Clara, CA, USA). The column was equilibrated with 2% acetonitrile for 3 min. Elution was performed at a flow rate of 0.2 mL/min and a column temperature of 25 °C in a solution of 0.15 M of NH_4_ acetate with 0.1% (*v*/*v*) HCOOH with a linear gradient of acetonitrile from 2% to 16% over 5 min and then at 50% for 1.5 min. The chromatograms were processed using MassLynx 4.1 (Waters, Milford, MA, USA). The data were collected and analyzed using IBM SPSS Statistics (v. 22; IBM SPSS, Armonk, NY, USA).

### 2.7. Statistical Analysis

Where appropriate, the data are presented as the mean ± standard deviation (SD). A comparison of group dispersions (homoscedasticity) was performed using the Fisher–Snedecor test with a significance threshold of α = 0.05. Paired and unpaired *t*-tests were conducted to compare the results between the experimental groups. For the analysis, 2-sided *p* < 0.05 was considered to be significant.

## 3. Results

### 3.1. nPt Characteristics

The nPts were assessed using dynamic light scattering and transmission electron microscopy. The formation of the face-centered cubic phase of the metal was confirmed by the electron diffraction pattern according to the ratios of diffraction ring diameters. The ratios of diffraction ring diameters corresponding to {111}, {200}, {220}, {311}, and {222} planes were √3:, √4:, √8:, √11:, and √12, respectively. The diffuseness of the diffraction rings was due to the small particle size. The absence of extraneous reflexes indicated the absence of an admixture phase of oxides, hydroxides, or complex metal compounds (Figure 2A). The nPt size determined via transmission electron microscopy was 3 ± 1 nm. Dynamic light scattering was used to determine the hydrodynamic particle size, which was 28 nm (Figure 2B), while the ζ potential of the nanoparticles was negative at −8.83 mV (Figure 2C). The hydrodynamic particle size determined by dynamic light scattering exceeded the nanoparticle diameter determined by transmission electron microscopy. This could be because the PVP stabilizer shell thickness contributed to the hydrodynamic particle size, because of the possible formation of associate nanoparticles, or because of features of the hydrodynamic size calculation using dynamic light scattering. The nPt preparations obtained were suitable for further study of their biological effects because they did not contain impurities from the reaction mixture and did not form aggregates in the saline solution.

### 3.2. Morphological Examination of the Brain

The sham operation was accompanied by a background level of hyperchromic neurons (~5%) in the cornu ammonis (CA1 region) of the hippocampus (Figure 3A,B). This indicator increased to 22 ± 7% three days after tBCAO in the rats injected with 0.9% NaCl, but the level of hyperchromic neurons did not exceed the background level (2.5 ± 1.2%) in the rats injected with nPts. However, one week after tBCAO, approximately the same high rate of neuronal apoptosis (~22%) was observed in both groups, as shown in Figure 3.

### 3.3. Influence of nPts on CBF and MAP

The introduction of nPts per se had almost no effect on the MAP, but it slightly increased the level of CBF in the intact rats (Figure 4A,B). The tBCAO caused an approximately twofold decrease in the MAP and a significant decrease in the CBF from 23.4 to 3.3–3.4 perfusion units (*p* < 0.001). After 2 h of reperfusion, the MAP level in the tBCAO and tBCAO + nPt groups remained significantly lower than the initial level (*p* < 0.01), while the CBF level in the tBCAO group was approximately 60% of the initial level (*p* < 0.001), which indicated cerebral hypoperfusion. At the same time point, this indicator was significantly higher (*p* < 0.001) and was more than 95% of the initial level in the tBCAO + nPt group. Figure 4C,D show the changes in CBF in these groups of animals.

### 3.4. Determination of Pt in Rat Tissues

A high level of Pt was observed in the liver, and a significantly lower level was observed in the blood three hours after the administration of nPt (Table 1). In the hippocampus, the Pt level could not be determined in some animals because it was below the detection limit (0.3 ng/g of sample). In other rats, the Pt content in the hippocampus did not exceed 1.5 ng/g. Taking into account the average volume of blood in the hippocampus (7.15 µL/g [42]) and assuming a blood density of 1.05 g/mL [43], we calculated that the content of Pt directly in the nervous tissue of the hippocampus (i.e., without blood) averaged less than 1 ng/g (Table 1).

### 3.5. Effect of nPts on the Brain and Plasma Aminothiols

The tBCAO caused a drop in the level of rCys and especially in the level of rGSH (see Figure 5). Interestingly, the level of rHcy, in contrast, was increased. The introduction of nPts per se caused an increase in the rCys level with no effect on rGSH and rHcy. In the presence of tBCAO, the introduction of nPt prevented a decline in rCys and rGSH, but it had almost no effect on the rHcy level.

Changes in the global methylation potential (SAM/SAH) in the hippocampal tissue are shown in Figure 6. Despite a fairly significant variation in this indicator in the control groups, there was a significant decrease in its value three days after tBCAO in the rats that were injected with 0.9% NaCl. However, in the rats injected with nPts, the global methylation potential remained at the same level.

A decrease in the level of reduced GSH was already observed after three hours of reperfusion in the hippocampus in both the 0.9% NaCl- and nPt-treated rats (Figure 7A). This indicator continued to decrease three days after tBCAO in the NaCl-treated group, but it returned to the initial level in the nPt-treated group (Figure 7A). The level of oxidized GSH increased significantly after three hours of reperfusion in all groups with tBCAO (Figure 7B). In general, it could be conjectured that tBCAO caused a significant decrease in the GSH redox status (i.e., reduced/oxidized GSH ratio) in the hippocampus, but nPts attenuated this effect in the first hours of reperfusion (Figure 7C).

## 4. Discussion

### 4.1. nPt Synthesis

Aqueous dispersions of nPts can be obtained using a variety of chemical methods based on the reduction of Pt salts in aqueous solutions with various reducing agents in the presence of stabilizers [44]. However, not all nPt drugs are suitable for studying the bioeffects of nPts because of the presence of nonreduced metal ions, toxic stabilizers, or oxidation products of the reducing agent used, which are present in excess. In addition, nPts stabilized by ionic molecules can exhibit aggregation in salt media [16].

We synthesized nPts by reducing an aqueous solution of Pt salts with NaBH_4_, which can quickly and completely restore Pt ions [45]. It is important to note that the obtained nPts were characterized by small sizes (3 ± 1 nm), which could play a positive role in their ability to penetrate the BBB. We used PVP as a stabilizer because it is nontoxic and biocompatible. nPts were separated from the reaction mixture using ultracentrifugation, followed by resuspension in saline. Thus, we obtained a stable nPt formulation containing no impurities from the reaction mixture, which was suitable for biomedical applications.

### 4.2. The Effect of nPts on Aminothiols

The effect of nPts is associated with the suppression of ROS production (particularly that of O_2_^−^) in cerebral tissues [33,34]. Aminothiols are one of the natural components of antioxidant protection, but they undergo oxidation by ROS and their oxidation products have a damaging effect on both the vascular endothelium and neurons. The level of their reduced forms is low, and in rats, their level usually constitutes 5–10% of the total content in blood plasma. Reduced GSH is a major thiol inside cells, and its content is many times higher than that of other thiols. Ischemia or brain trauma causes systemic disturbances in metabolism involving the peripheral vessels that induce a massive release of ROS, particularly of superoxide anion, which is apparently the result of activation of the sympathoadrenal system [7,8]. This is the most obvious reason for the rapid decrease in the level of reduced aminothiols and in the level of total blood plasma sulfhydryl groups [19,46]. However, there is still no clear understanding of the role of systemic OS in the pathogenesis of ischemic brain damage and the prospects for using systemic OS as a therapeutic target in stroke.

In our experiments, the introduction of nPts prevented the decrease in the levels of rCys and rGSH during the first hours of reperfusion, which demonstrated their effectiveness in suppressing systemic OS. It has been shown that the administration of antioxidant and nonspecific carvedilol adrenergic antagonists significantly increases the levels of rCys, rGSH, and rHcy in plasma in intact rats but is unable to maintain their levels in the tBCAO model, despite their pronounced antioxidant effect in brain tissue [47,48]. However, the β-adrenergic receptor antagonist metoprolol (which is not an antioxidant in high doses) attenuates the postischemic decrease in rCys, rGSH, and rHcy in the same model [20].

GSH is the main intracellular antioxidant, and it is used to neutralize lipid hydroperoxides and H_2_O_2_ (via GSH-peroxidase) and remove xenobiotics from cells (via GSH-S-transferase); maintaining its level prevents pro-inflammatory pathways, such as activation of c-Jun N-terminal kinase [49]. A decrease in the level of reduced GSH in the brain tissue is a consequence of the development of local OS in the brain during ischemia/reperfusion injury. Although we did not find a protective effect of nPts on the content of the reduced form of GSH directly in the hippocampus in the first hours after tBCAO, the administration of nPts attenuated the growth in the oxidized form of GSH, and therefore the decrease in its redox status was less pronounced during this period. The positive effect of a single injection of nPts on GSH synthesis persisted even three days after reperfusion, which was expressed in the ability of the brain to maintain a normal level of reduced GSH despite an increase in its oxidized form and, as a result, a violation of its redox status.

The SAM/SAH ratio (global methylation index) is also an important indicator of cell viability, which decreases with ischemic brain damage [20]. Our results showed that nPts suppress this ischemia–reperfusion effect as well. Thus, it can be concluded that nPts have a protective effect on both the general and brain metabolism of aminothiols, which is consistent with their antioxidant activity in vivo.

### 4.3. The Effect of nPts on CBF

The model of ischemia we used leads to a temporary decrease in cerebral flow to critical thresholds at which both necrosis and apoptosis processes are triggered. Therefore, a rapid restoration of blood flow is very important for the normal activity of neurons. The return of spontaneous circulation does not naturally result in a recovery of cerebral perfusion, as cerebral perfusion failure after ischemia is well described in animal models with no reflow, hypoperfusion, and hyperperfusion [50]. Hyperperfusion is the result of a number of interrelated factors: impaired metabolism of vasodilators (nitric oxide—NO and prostacyclin), an increase in the level of vasoconstrictors (endothelin-1 and noradrenaline), a decrease in the production of anticoagulants by the endothelium with the formation of microthrombi, cerebral edema at the stage of ischemia because of hemostasis, and an increase in blood–brain barrier permeability. ROS and reactive nitrogen species formed during the activation of inducible NO synthase (iNOS), NADPH oxidase, and cyclooxygenase are actively involved in these processes [11,51]. Superoxide anion produced during the last two reactions in both cerebral vessels [52] and peripheral arterioles [37] reacts with NO to become peroxynitrile, which in turn inhibits Ca + 2/K + BK channels and leads to vasoconstriction [53].

Because the ability of nPts to reduce superoxide anion production has been shown in both cell cultures and models of cerebral ischemia [16,33], the effectiveness of nPts in regulating cerebrovascular autoregulation, as shown in the present study, may be the result of a direct antioxidant effect of nPts on brain vessel endothelium. Although little is known about the specific mechanisms of ED attenuation by nanoparticles in vivo, experimental work in cell cultures has shown that nPts can maintain NO bioavailability not only by neutralizing ROS but also by catalyzing the release of NO from its bound forms [54] and preventing iNOS activation [55,56].

In addition, CBF is under the neurogenic influence of sympathetic fibers and signals associated with the intensity of neuron metabolism [57]. Although the data reporting an influence of the sympathetic nervous system on CBF are controversial [58,59], the role of spreading depolarization of gray matter in the brain during CBF disruption is undeniable and is considered to be a therapeutic target for secondary lesions in the brain [11]. Spreading depolarization causes not only pronounced local metabolic effects but also vasospasm in the brain, which contributes to enlargement of the ischemic area (penumbra) during the reperfusion period [60]. Therefore, the elimination of this ischemic hypoperfusion effect in the presence of nPts might also be explained through indirect mechanisms involving suppression of sympathetic nervous tissue activation or spreading depolarization.

### 4.4. Protective Action of nPts on Hippocampal Neurons during tBCAO

It is known that hippocampal structures are highly vulnerable to ischemia, and the analysis of regions such as the cornu ammonis or dentate gyrus is widely used to assess brain damage resulting from global ischemia [61]. Apoptotic changes in the hippocampus develop for quite a long time after ischemia/reperfusion; therefore, this process becomes visible morphologically only after a few days, and within 7–10 days, a stable pattern of hippocampal damage is usually formed in models of global cerebral ischemia in rats [62,63]. In this regard, we sought to determine the intensity of apoptosis approximately at the peak of its growth (three days) and by the time a stable focus of damage to the hippocampus was formed (seven days). A previous study demonstrated the protective effect of nPt on the volume of infarction in the brain cortex (but not in the dorsal striatum) in a more severe model of ischemia (1 h occlusion of the mesencephalic artery). Our results demonstrated that nPts effectively prevented neuronal death in the hippocampus during tBCAO; however, their effect was not long-lasting after a single dose.

Based on the numerous studies that demonstrated the direct antioxidant action of nPts in vitro and in vivo, the preserved viability of neurons in the hippocampus in the tBCAO model could also be explained by an antioxidant effect of nPts directly on the brain tissue. In addition, these studies demonstrated a high rate of nPt absorption and distribution in the body. One of the most urgent problems of using various nanoparticles in stroke therapy is their poor distribution in the brain [14]. Once inside a cow, nPts are rapidly absorbed by the liver, kidneys, and spleen by monocytes; therefore, their half-life in blood circulation is severely limited. The endothelium is also actively involved in the transport of nPts from the blood to tissues, but to date, there have been few studies on the ability of nPts to overcome the BBB in normal and pathological conditions. In general, it can be said that there is an inverse relationship between the sizes of nanoparticles of various origins and the ability to penetrate the BBB [14]. It is also known that various damaging factors (trauma, ischemia, and hypoxia) cause an increase in the BBB when the brain is most susceptible to damage [64]. Therefore, in order to have an idea of the actual distribution of nPts in the body (especially in the brain) during the direct action of the damaging factor, we determined the Pt content three hours after the start of reperfusion. None of the previous studies measured the level of nPts in the brain tissue, and our study showed that after intraperitoneal administration, nPts rapidly accumulated in the liver, where their level could reach hundreds of nanograms per gram, but in the hippocampus itself, their content was less than 1 ng/g; i.e., their accumulation was not observed in the most sensitive area of the brain to ischemia/reperfusion. ICP-MS is quite sensitive (detection limit of 0.3 ng/g of sample), but in a third of the animals, the nPt level in the hippocampus was below this threshold.

In various in vitro studies, the biological effects of nPts have been identified in the concentration range of 0.5–200 μg/mL [65,66,67,68,69], which is more than three orders of magnitude higher than their level in the rat hippocampus determined in the present study. This raises questions about whether such a level of nPts is sufficient for effective decomposition of ROS in the tissue or whether the protective effect of nPts is due to their indirect effect on nervous tissues and peripheral organs as well as the sympathetic–adrenal system. It is known that platinum nanoparticles not only have antioxidant activity but also pro-oxidant activity; for example, platinum nanoparticles are mimetics of peroxidase and catecholoxidase [70]. Bioactive catechols include epinephrine, norepinephrine, and dopamine, while it has been shown that platinum nanoparticles can accelerate the oxidation of dopamine [71]. Thus, it is possible that the inhibition of systemic OS and postischemic hypoperfusion is due to accelerated oxidation of bioactive catechols on the surface of nPts. In this regard, it becomes interesting to study the possibility of nPt accumulation in the peripheral sympathetic nerves and its effect on presynaptic β1-adrenergic receptors, which play a key role in the activation of this system [7]. However, little is known about the molecular targets of nPts in endothelial cells and neurons.

Our study had several limitations. First, our focus was only on the effect of nPts on apoptosis, CBF, and the aminothiols system, leaving aside other important parameters such as necrosis, the ROS itself, catecholamines, inflammatory mediators, and ED. Second, we only studied the acute response to tBCAO and not the long-term or medium-term consequences of nPt accumulation in organs. Although nPts are considered nontoxic, they can show cardiotoxicity at high doses [72]. Finally, we studied only a single administration of nPts but not the effects of its regular administration.

## 5. Conclusions

In the present study, we synthesized and purified nPts to demonstrate their effectiveness as neuro- and vasoprotective agents in a tBCAO model in rats. The tBCAO model was not complicated by the presence of products resulting from incomplete Pt reduction. We found that purified nPts prevented the apoptotic death of hippocampal neurons, eliminated the effect of postischemic hypoperfusion, and prevented the decline in the levels of reduced aminothiols in blood plasma. Moreover, nPts attenuated the decline in the GSH redox status and the global methylation index directly in the brain, which indicated the ability of nPts to suppress the development of both local and systemic OS. However, the level of nPts in the brain was significantly lower than that at which the direct antioxidant effects of nanoparticles are observed. This suggests firstly that the mechanism of neuroprotective action of nPt is indirect; secondly, it points to the important role of extracerebral mechanisms in the development of secondary brain damage. Therefore, nPts can be a very useful tool for both therapy and the search for new targets for the treatment of cerebrovascular diseases.

## Figures and Tables

**Figure 1 jfb-14-00348-f001:**
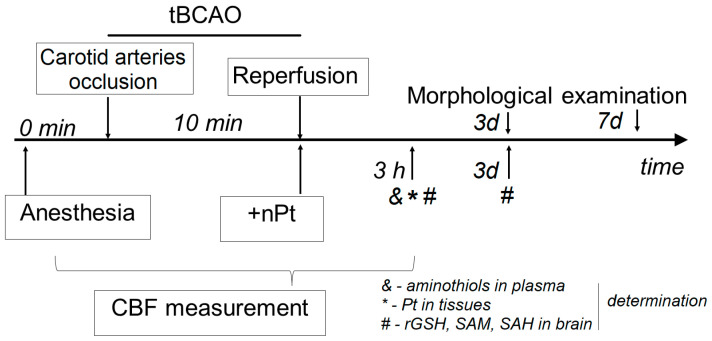
Outline of the induction of tBCAO and measurements taken.

**Figure 2 jfb-14-00348-f002:**
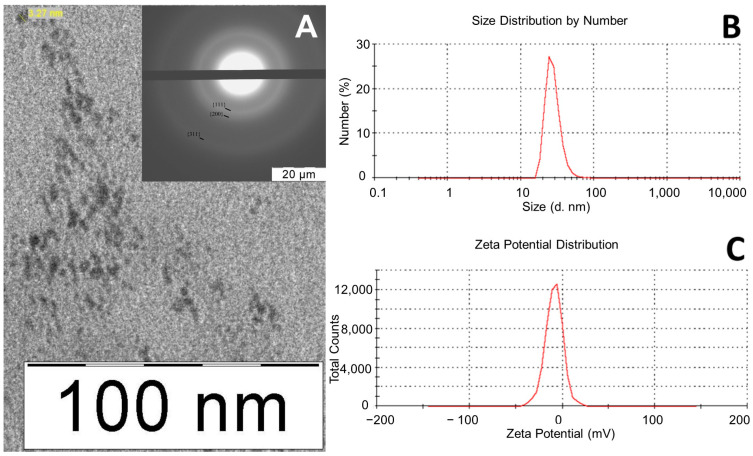
(**A**) Electron micrograph of nPts. The inset shows the electron diffraction pattern. (**B**) Distribution of nPts according to hydrodynamic size as determined using dynamic light scattering. (**C**) nPt distribution according to ζ potential.

**Figure 3 jfb-14-00348-f003:**
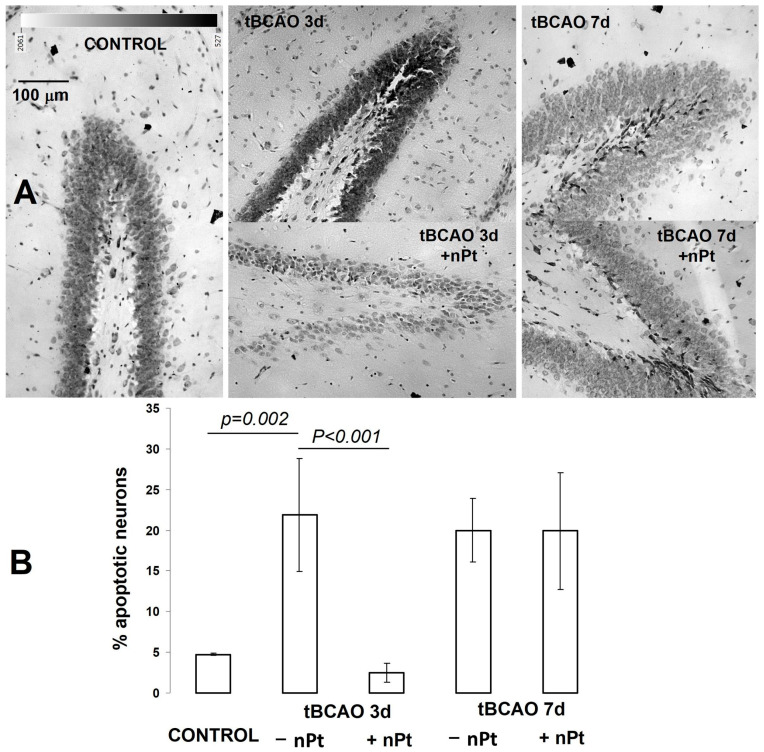
(**A**) Images of rat cornu ammonis. (**B**) Level of hyperchromic neurons in the rat cornu ammonis (CA1 region) of the hippocampus after tBCAO (*n* = 5).

**Figure 4 jfb-14-00348-f004:**
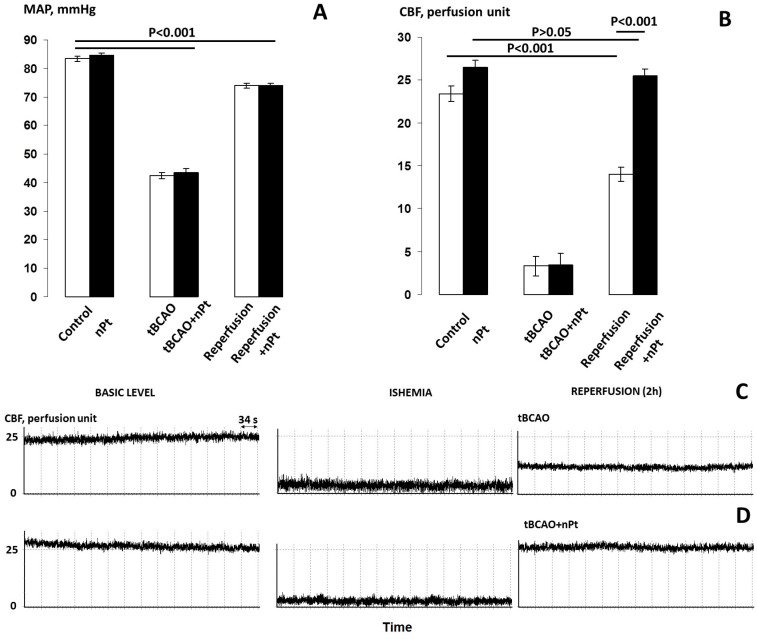
(**A**,**B**) Comparison of MAP and CBF, respectively, in different experimental groups. (**C**,**D**) Typical CBF curves for the tBCAO and tBCAO + nPt groups, respectively.

**Figure 5 jfb-14-00348-f005:**
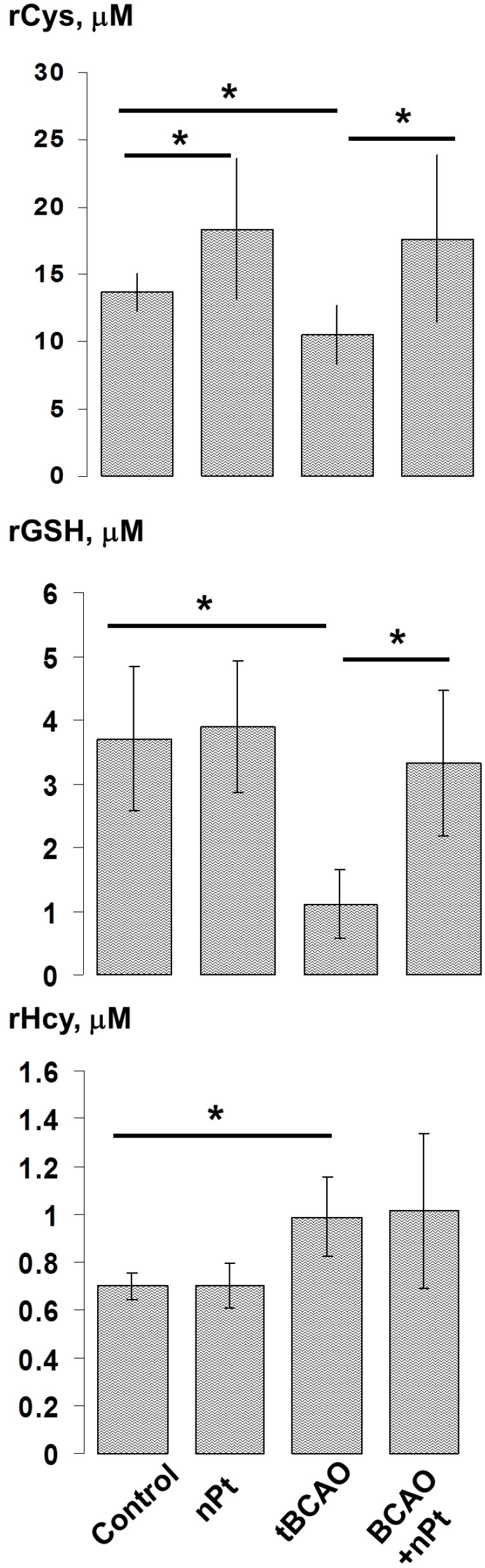
The effect of tBCAO and nPts on the plasma levels of reduced thiols. * *p* < 0.05.

**Figure 6 jfb-14-00348-f006:**
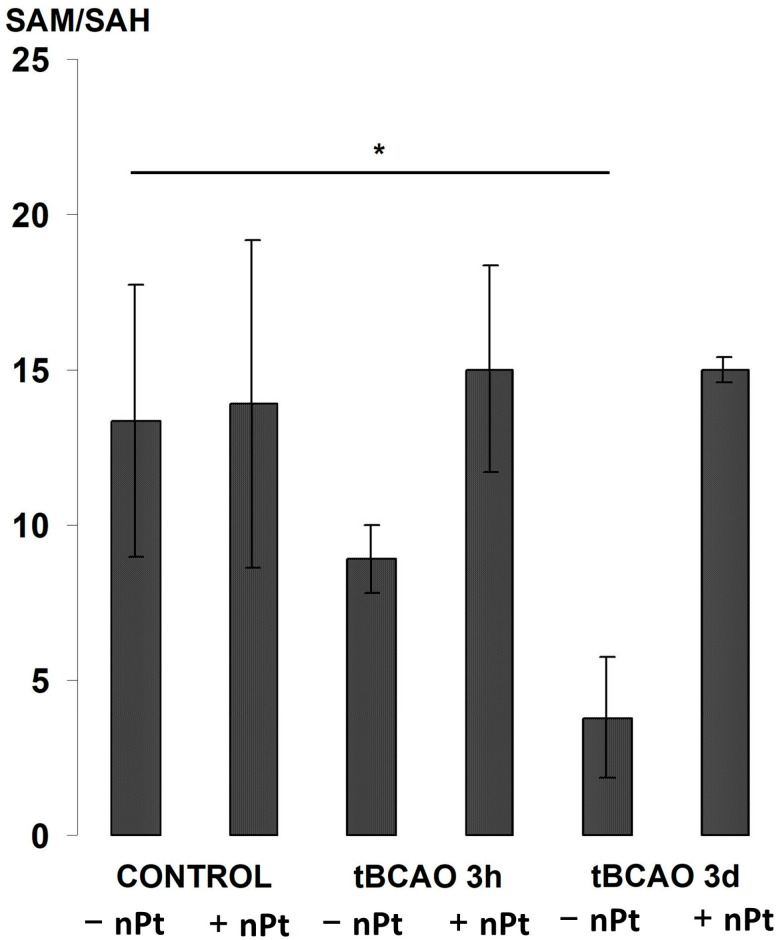
The effect of tBCAO and nPts on SAM/SAH ratio in the hippocampus. * *p* < 0.05.

**Figure 7 jfb-14-00348-f007:**
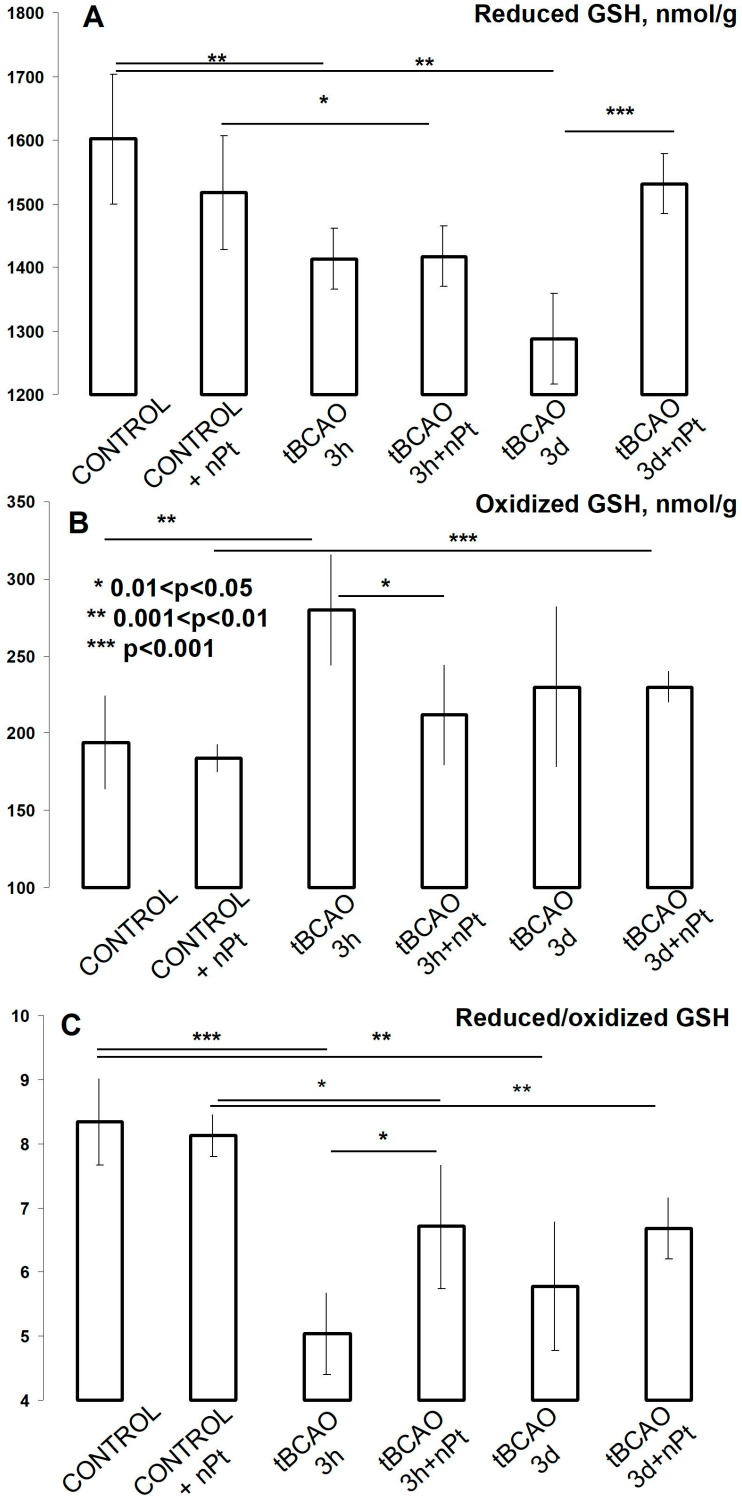
The effect of tBCAO and nPts on the levels of reduced (**A**) and oxidized (**B**) GSH and their ratio (**C**) in the hippocampus. * *p* < 0.05, ** *p* < 0.01, and *** *p* < 0.001.

**Table 1 jfb-14-00348-t001:** Pt levels (ng/g) in rat tissues (*N* = 9).

Rat No.	Tissue Type			
	Liver	Blood	Hippocampus (Whole)	Hippocampus (without Blood)
Median	167	44	0.83	0.74
Min.	26	2.9	<0.3 (*N* = 3)	<0.3 (*N* = 3)
Max.	467	97	1.5	1.1

## Data Availability

The data presented in this study are available from the corresponding author upon request.

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
