# Peer review of "Neuroprotective Effect of Platinum Nanoparticles Is Not Associated with Their Accumulation in the Brain of Rats"

_jfb, 2023, doi:10.3390/jfb14070348_

Round 1
Reviewer 1 Report
The study performed by authors “Neuroprotective Effect of Platinum Nanoparticles is not Associated with Their Accumulation in the Brain of Rats”, This work showed some interesting information. there are some minor points to be further addressed. Mostly linked to the research topic and experimental results. However, the manuscript needs minor revision and following questions/points should be further clarified
Some questions and comments as follow:
Abstract:
1. The weakness of this abstract is that it does not provide much information about the methods and results of the study. For example, it is unclear how many animals were used in the study, what specific measurements were taken to assess brain function and damage. Furthermore, the abstract does not provide any information about the potential limitations or implications of the findings, such as how they may be relevant to human health or future research in this area.
Introduction:
2. while it provides a lot of information about the background and previous studies on the topic, it does not clearly state the objectives or hypotheses of the current study. Finally, it may benefit from a more concise and clear writing style to improve readability.
3. How do nanoparticles of noble metals, such as platinum, gold, titanium, and palladium, play a role in the treatment and diagnosis of cerebrovascular diseases?
4. how has nanotechnology influenced the treatment and diagnosis of cerebrovascular diseases, and what are some examples of biocompatible nanoparticles used for this purpose?
5. What role does oxidative stress play in the pathogenesis of stroke or brain injury, and how might the use of low-molecular-weight aminothiols and nPt help to mitigate these effects
Materials and Methods
6. Lack of detailed information: The section does not provide full details about some procedures, such as the method used to determine the hydrodynamic size and zeta potential of the particles.
7. Unclear information: Some parts of the section are unclear, for example, it is not clear why five rats from each group were euthanized for morphological examination of the brain after 3 and 7 days after tBCAO and why nine rats from the third group were euthanized for determination of nPt in hippocampus, blood, and liver after 3h of reperfusion.
8.Inadequate information: The section does not provide sufficient information about the instruments used for analysis, such as the brand and model of the high-performance liquid chromatography (HPLC) system.
Results:
9. some of the findings are not clearly linked to the research question or hypothesis, leaving the reader unclear about their relevance. Further, the statistical analysis used to support the findings is not described in detail, which can lead to questions regarding validity and reliability.
Discussion:
10. The weakness of the discussion section is that it mainly presents the findings and results of the study without delving deeper into the implications and limitations of the research.
The section could benefit from a more critical analysis of the results, including a discussion of potential confounding factors, alternative explanations for the observed effects, and the generalizability of the findings beyond the specific experimental conditions. Additionally, the section could benefit from suggestions for future research directions based on the current findings.
11. What is the process for synthesizing stably aggregated nPt aqueous dispersions and why is it important for biomedical applications?
12. How does nPt affect the levels of aminothiols in the brain during ischemia-reperfusion injury, and what implications does this have for its antioxidant activity in vivo?
13. What is the role of nPt in regulating cerebral blood flow during and after ischemic events, and how does it potentially protect hippocampal neurons from damage?
Minor editing of English language required
Author Response
Dear reviewer, we thank you for your attention to our research and valuable recommendations, following which we revised the manuscript. Below we provide answers to your comments, and also attach the manuscript file, where the changes corresponding to the new version are highlighted in red.
Some questions and comments as follow:
- Abstract: The weakness of this abstract is that it does not provide much information about the methods and results of the study. For example, it is unclear how many animals were used in the study, what specific measurements were taken to assess brain function and damage. Furthermore, the abstract does not provide any information about the potential limitations or implications of the findings, such as how they may be relevant to human health or future research in this area.
Ans.: addition of the abstract in accordance with the commentary of the reviewer is highly desirable, but the volume of the abstract is strictly limited by the rules of the journal. We have tried to meet these requirements to the best of our ability and have revised this section: “Platinum nanoparticles (nPt) have neuroprotective/antioxidant properties, but the mechanisms of their action in cerebrovascular disease remain unclear. We investigated the brain bioavailability of nPt and their effects on brain damage, cerebral blood flow (CBF), and development of brain and systemic oxidative stress (OS) in a model of cerebral ischemia (hemorrhage + temporary bilateral common carotid artery occlusion, tBCAO) in rats. nPt (0.04 g/l, 3 ± 1 nm diameter) was administered to rats (N=19) intraperitoneally at the start of blood reperfusion. Measurement of CBF via laser Doppler flowmetry revealed that nPt caused a rapid attenuation of postischemic hypoperfusion. nPt attenuated the apoptosis of hippocampal neurons, the decrease in reduced aminothiols level in plasma, and the glutathione redox status in the brain, which were induced by tBCAO. The content of Pt in the brain was extremely low (≤ 1 ng/g). Thus, nPt, despite the extremely low brain bioavailability, can attenuate the development of brain OS, CBF dysregulation, and neuronal apoptosis. This may indicate that the neuroprotective effects of nPt are due to indirect mechanisms rather than direct activity in the brain tissue. Research on such mechanisms may offer a promising trend in the treatment of acute disorders of CBF.” - Introduction:
while it provides a lot of information about the background and previous studies on the topic, it does not clearly state the objectives or hypotheses of the current study. Finally, it may benefit from a more concise and clear writing style to improve readability.
Ans.: this section has been revised in accordance with the reviewer's comment. We have added a number of propositions to support our hypothesis. First, these are the basic concepts of therapy for acute disorders of cerebral blood flow: “To date, there are essentially two concepts for the treatment of acute CBF disorders. The first involves the use of thrombolytics (recombinant tissue plasminogen activator) or endovascular thrombectomy, which enables recanalization and restoration of blood flow in damaged vessels. The effectiveness of this approach in practice is significantly limited by the time window available (up to 3-4.5 hours for thrombolysis and 6-24 hours for endovascular thrombectomy), and the possibility of severe complications or contraindications after use [5,6]. The second concept involves the use of various agents that have a neuroprotective effect.
The brain has a high rate of oxygen consumption, and unlike most organs, it has no internal energy reserves. This causes a rapid development of oxidative stress (OS) in the brain, a process in which the generation of reactive oxygen species (ROS) prevails over the ability to eliminate them, both under conditions of ischemia and during subsequent reperfusion. Also, cerebral ischemia causes rapid and strong activation of the sympathetic nervous system [7], which, apparently, triggers the development of systemic endothelial dysfunction (ED) [8]. Due to the key role of OS in the development of ischemic brain damage [9], ED [10], and postischemic hypoperfusion [11], the use of antioxidants is one of the promising areas of neuroprotection. For example, there have already been a few clinical studies showing the effectiveness of the low-molecular-weight antioxidants edaravone [12] and N-acetylcysteine [13].”
Second, we described in more detail the role of aminothiols as markers of cerebral ischemia: “Low-molecular-weight aminothiols (cysteine—Cys, glutathione—GSH, homocysteine—Hcy, and others) are highly sensitive to OS. In response to cerebral ischemia, there is a rapid decrease in the content of their reduced forms in blood plasma, as well as a drop in the redox status of the main intracellular antioxidant, GSH, in the nervous tissue itself [19,20]. GSH has previously been shown to be associated with stroke severity [21]. The important protective role of GSH is also confirmed by the efficacy of using N-acetylcysteine (a readily available substrate for GSH synthesis) in models of ischemia [22,23] and in the treatment of stroke [13]. An elevated Hcy level, which damaging effect is associated with the induction of OS [24], is considered a risk factor for stroke since a decrease in Hcy level can reduce this risk [25]. The effect of nPt on the homeostasis of aminothiols in cells is still poorly understood. Previously, it was shown that nPt decreased the cellular GSH level, and this effect correlated with the particle size in an inverse manner but appeared not to be based on the formation of ROS [26]. To the best of our knowledge, studies on the effect of nPt on aminothiols under stress conditions have not yet been carried out.
Under ischemia–reperfusion injury, the development of OS leads to the activation of a number of transmethylases in the nervous tissue, which manifests as hypermethylation of DNA, non-coding RNA, and histones, and disruption of the biosynthesis of polyamines and acetylcholine, which, in turn, enhance inflammatory damage to neurons [27]. During these transmethylation reactions, S-adenosylmethionine (SAM) is consumed and S-adenosylhomocysteine (SAH) is formed. Thus, the SAM/SAH ratio is referred to as the global methylation index [28]. Numerous works involving models of cerebral ischemia have shown a protective effect of SAM administration [29-31], including decreasing the GSH content in the brain [32]. But studies on the effect of nPt on the balance of SAM and SAH have not yet been conducted.
Thirdly, we also mentioned the problem of bioavailability of neuroprotectors, which will play an important role in the manuscript: “In the study on the neuroprotective effect of substances, one of the key issues is their bioavailability, i.e., the ability to penetrate through the blood–brain barrier (BBB), which is a complex of membranes and intercellular contacts formed by the endothelium of blood vessels, astrocytes, and pericytes [35]. The BBB prevents the transfer of many substances from the blood to the brain. Although it has previously been shown that metal nanoparticles, in particular gold, penetrate the BBB through the mechanisms of passive diffusion, carrier-mediated transport, adsorptive-mediated endocytosis, or pinocytosis [36], there are no data yet on how the BBB can be made permeable to nPt.”
The formulation of the hypothesis has undergone correction: “Thus, based on the neuroprotective effect of nPt in vivo, we set out to shed light on 1) whether nPt is able to effectively suppress systemic OS and postischemic hypoperfusion induced by acute impairment of cerebral blood flow, and 2) whether the effect of nPt is due to a direct effect on brain tissue or due to the extracerebral effect of nanoparticles. To achieve these objectives, we determined indicators of systemic (reduced plasma levels of aminothiols) and brain (reduced GSH, SAM, and SAH in the hippocampus) OS and CBF and brain nPt content using a model of temporary global cerebral ischemia in rats.” - How do nanoparticles of noble metals, such as platinum, gold, titanium, and palladium, play a role in the treatment and diagnosis of cerebrovascular diseases?
Ans.: Platinum nanoparticles with antioxidant properties are promising for the treatment of ischemic stroke (Takamiya, M. et al. J. Neurosci. Res. 2011, 89, 125-133. Takamiya, M. et al. Neuroscience, 2012, 221, 47-55.), Parkinson's disease (Nellore, J. et al. Journal of Neurodegenerative Diseases, 2013, 972391.). Gold nanoparticles are used in analytical systems for measuring biomarkers of cardiovascular diseases. It is possible to use metal nanoparticles as contrasting agents in imaging methods. (Xu, H. et al. Sig Transduct Target Ther. 2022, 7, 231). - how has nanotechnology influenced the treatment and diagnosis of cerebrovascular diseases, and what are some examples of biocompatible nanoparticles used for this purpose?
Ans.: The small size of nanoparticles facilitates their penetration through the blood-brain barrier and makes it possible to use them for visualization of amyloid deposits in the brain, delivery of radiosensitizers or small interfering RNAs as part of organic nanoparticles for glioblastoma therapy. Cerium dioxide nanoparticles with antioxidant properties can reduce the effects of traumatic brain injury and facilitate the course of amyotrophic lateral sclerosis, and fullerene nanoparticles with antioxidant properties weaken the effects of ischemic stroke. Polydopamine nanoparticles with metformin or narigenin nanoparticles with vitamin E with antioxidant properties are promising in Parkinson's disease. The loading of anticonvulsants and antioxidants into organic nanoparticles for the treatment of epilepsy is promising.(Ngowi, EE et al. Front. Bioeng. Biotechnol. 2021, 9, 629832, Luo, Y et al. Part. Syst. Charact. 2021, 38, 2000311). Promising in the treatment of cerebrovascular and cardiovascular diseases is the use of organic nanoparticles with a variety of antioxidants - TEMPO, ferulic acid, vanillyl alcohol, SOD, as well as the use of organic nanoparticles loaded with small interfering RNAs to stop the progress of atherosclerosis or reduce the necrosis zone in stroke. (Xu, H et al. Sig Transduct Target Ther. 2022, 7, 231.) - What role does oxidative stress play in the pathogenesis of stroke or brain injury, and how might the use of low-molecular-weight aminothiols and nPt help to mitigate these effects.
Ans.: oxidative stress play a key role in the pathogenesis of brain and vascular injury, however, we believe that it is not worth overloading the manuscript with an excessive amount of detail and therefore we provide links to recent reviews in the introduction section: “The brain has a high rate of oxygen consumption, and unlike most organs, it has no internal energy reserves. This causes a rapid development of oxidative stress (OS) in the brain, a process in which the generation of reactive oxygen species (ROS) prevails over the ability to eliminate them, both under conditions of ischemia and during subsequent reperfusion. Also, cerebral ischemia causes rapid and strong activation of the sympathetic nervous system [7], which, apparently, triggers the development of systemic endothelial dysfunction (ED) [8]. Due to the key role of OS in the development of ischemic brain damage [9], ED [10], and postischemic hypoperfusion [11], the use of antioxidants is one of the promising areas of neuroprotection. For example, there have already been a few clinical studies showing the effectiveness of the low-molecular-weight antioxidants edaravone [12] and N-acetylcysteine [13]”. In our opinion, it is important here not to be limited to OS processes directly in the brain, although they certainly play a primary role, but also to explore the potential possibilities of suppressing systemic OS in peripheral organs.
Also, this section was supplemented by the role of aminothiols as markers of acute cerebral ischemia and targets of metabolic therapy for stroke: “Low-molecular-weight aminothiols (cysteine—Cys, glutathione—GSH, homocysteine—Hcy, and others) are highly sensitive to OS. In response to cerebral ischemia, there is a rapid decrease in the content of their reduced forms in blood plasma, as well as a drop in the redox status of the main intracellular antioxidant, GSH, in the nervous tissue itself [19,20]. GSH has previously been shown to be associated with stroke severity [21]. The important protective role of GSH is also confirmed by the efficacy of using N-acetylcysteine (a readily available substrate for GSH synthesis) in models of ischemia [22,23] and in the treatment of stroke [13]. An elevated Hcy level, which damaging effect is associated with the induction of OS [24], is considered a risk factor for stroke since a decrease in Hcy level can reduce this risk [25]. The effect of nPt on the homeostasis of aminothiols in cells is still poorly understood. Previously, it was shown that nPt decreased the cellular GSH level, and this effect correlated with the particle size in an inverse manner but appeared not to be based on the formation of ROS [26]. To the best of our knowledge, studies on the effect of nPt on aminothiols under stress conditions have not yet been carried out.”
- Materials and Methods
Lack of detailed information: The section does not provide full details about some procedures, such as the method used to determine the hydrodynamic size and zeta potential of the particles.
Ans.: corresponding text was added to 2.3 section: “A photonic particle analyzer Zetasizer Nano ZS (Malvern, UK) was used to determine the size and zeta potential of the particles. The analyzer has a particle measurement range from 0.6 to 6000 nm. The operating temperature range is 2°C – 120°C, the angle of detection of scattered light is 173°C, a helium-neon laser with a wavelength of 633 nm is used as a light source, the power of the light source is 5 MW. The device determines particle sizes by measuring the rate of fluctuation of scattered light by particles. The measurement was carried out in automatic mode according to the standard procedure. The glass cuvette was filled with 1 ml of the sample, loaded into the cuvette compartment of the device. The beam of light emitted by the laser passes through the attenuator and enters the sample cell. The light scattered by the particles is detected by the detector. The electrical signal of the detector, proportional to the light intensity, is processed by the correlator according to mathematical algorithms embedded in the software. When determining the zeta potential, an immersion-type electrode is lowered into a cuvette filled with a sample. The sample is exposed to an electric field, and the electrophoretic mobility of particles in an electric field is used to calculate the zeta potential. The solvent was water.” - Unclear information: Some parts of the section are unclear, for example, it is not clear why five rats from each group were euthanized for morphological examination of the brain after 3 and 7 days after tBCAO and why nine rats from the third group were euthanized for determination of nPt in hippocampus, blood, and liver after 3h of reperfusion.
Ans.: we carried out a morphological study of the hippocampus, as the most sensitive brain structure to ischemia/reperfusion, in order to identify neuronal apoptosis in it (as an indicator of brain damage). Morphological changes in the brain (apoptosis, necrosis) develop over a fairly long time after ischemia/reperfusion, and therefore this process is observed only after a few days, and in 7-10 days, a stable morphological picture of hippocampal damage is usually formed on models of global brain ischemia in rats [Zeng YS, Xu ZC. Neurosci Res. 2000;37(2):113-25.; Martone ME, Hu BR, Ellisman MH. 2000;10(5):610-6.; Ding C, He Q, Li PA. Exp Neurol. 2004 ;188(2):421-9.]. In this regard, we sought to determine the intensity of apoptosis approximately at the peak of its growth (3 days) and by the time a stable focus of damage to the hippocampus was formed (7 days). Thus, the text has been added to the discussion section: “Apoptotic changes in the hippocampus develop for quite a long time after ischemia/reperfusion; therefore, this process becomes visible morphologically only after a few days, and within 7-10 days, a stable pattern of hippocampal damage is usually formed in models of global cerebral ischemia in rats [62,63]. In this regard, we sought to determine the intensity of apoptosis approximately at the peak of its growth (three days) and by the time a stable focus of damage to the hippocampus was formed (seven days).”.
At the same time, ischemia/reperfusion acts as a damaging factor for only a few hours. Therefore, in order to have an idea of the actual distribution of nPt in the body during the direct action of this damage factor, we determined the Pt content 3 hours after the start of reperfusion. Relevant text has been added to the discussion section: “ Therefore, in order to have an idea of the actual distribution of nPt in the body, especially in the brain, during the direct action of the damaging factor, we determined the Pt content three hours after the start of reperfusion.”. - Inadequate information: The section does not provide sufficient information about the instruments used for analysis, such as the brand and model of the high-performance liquid chromatography (HPLC) system.
Ans.: the necessary information has been included in the manuscript: “We used an UPLC H-class ACQUITY system (Waters, Milford, MA) with a PDAλ UV detector (λ = 330nm) to measure the concentrations of reduced low-molecular-weight thiols.” - Results:
some of the findings are not clearly linked to the research question or hypothesis, leaving the reader unclear about their relevance.
Ans.: we tried to take this into account and made appropriate changes to the Introduction and Discussion sections.
Further, the statistical analysis used to support the findings is not described in detail, which can lead to questions regarding validity and reliability.
Ans.: we have indicated the methods of statistical processing of the results used in section 2.7 “Statistical Analysis”. - Discussion:
The weakness of the discussion section is that it mainly presents the findings and results of the study without delving deeper into the implications and limitations of the research.
The section could benefit from a more critical analysis of the results, including a discussion of potential confounding factors, alternative explanations for the observed effects, and the generalizability of the findings beyond the specific experimental conditions. Additionally, the section could benefit from suggestions for future research directions based on the current findings.
Ans.: many additions have been made to this section, including we have indicated the limitations of this study: “Our study has several limitations. First, our focus was only on the effect of nPt on apoptosis, CBF, and the aminothiols system, leaving aside other important parameters, such as necrosis, ROS itself, catecholamines, inflammatory mediators, and ED. Second, we only studied the acute response to tBCAO, but not the long-term or medium-term consequences of nPt accumulation in organs. Although nPts are considered non-toxic, they can show cardiotoxicity at high doses [72]. Finally, we studied only a single administration of nPt, but not the effects of its regular administration.” - What is the process for synthesizing stably aggregated nPt aqueous dispersions and why is it important for biomedical applications?
Ans.: the term “stably aggregated” was incorrect, and we abandoned its use. Here, on the contrary, it was meant to obtain nPt in a stable state, which does not allow nanoparticles to freely aggregate with each other. Therefore, the following changes were made to the text (section 3.1): “The nPt preparations obtained were suitable for further study of their biological effects because they did not contain impurities from the reaction mixture and they did not form aggregates in saline solution.” - How does nPt affect the levels of aminothiols in the brain during ischemia-reperfusion injury, and what implications does this have for its antioxidant activity in vivo?
Ans.: a rapid and significant decline in the level of reduced thiols or their redox status, primarily glutathione, occurs due to oxidative stress, which induces cerebral ischemia. “ GSH is the main intracellular antioxidant, and it is used to neutralize lipid hydroperoxides and H2O2 (by GSH-peroxidase), remove xenobiotics from cells (by GSH-S-transferase); maintaining its level prevents pro-inflammatory pathways, such as activation of c-Jun N-terminal kinase [49]. The lowering in the level of reduced GSH in the brain tissue is a consequence of the development of local OS in the brain during ischemia/reperfusion injury. Although we did not find a protective effect of nPt on the content of the reduced form of GSH directly in the hippocampus in the first hours after tBCAO, the administration of nPt attenuated the growth of the oxidized form of GSH and, therefore, the decrease in its redox status was less pronounced during this period. The positive effect of a single injection of nPt on GSH synthesis persisted even three days after reperfusion, which was expressed in the ability of the brain to maintain a normal level of reduced GSH despite an increase in its oxidized form and, as a result, a violation of its redox status. ” (the text has been added to the discussion section). - What is the role of nPt in regulating cerebral blood flow during and after ischemic events, and how does it potentially protect hippocampal neurons from damage?
Ans.: Maintaining the tone of the cerebral vascular wall is important for the blood supply to the brain. Nitric oxide (NO) is an important regulator of vascular wall tone. Reactive oxygen species oxidize nitric oxide. Platinum nanoparticles neutralize reactive oxygen species and possibly maintain the concentration of nitric oxide. Also the following text has been added to the discussion section: “Although little is known about the specific mechanisms of ED attenuation by nanoparticles in vivo, experimental work in cell cultures has shown that nPt can maintain NO bioavailability not only by neutralizing ROS but also by catalyzing the release of NO from its bound forms [54] and preventing of iNOS activation [55,56].”
Minor editing of English language required
Ans.: the manuscript has been subjected to language correction to eliminate errors.

Reviewer 2 Report
1. The size and shape of nanoparticles play an important role in biocompatibility. TEM data shown in Figure 2A don't provide any valuable information about the size and shape of nanoparticles which is of poor resolution. Authors need to provide better resolution TEM as well as HR-TEM need to be provided.
2. Purity of the nanoparticle is not confirmed. Authors may provide XRD and EDX to confirm the purity of the nanoparticles.
3. How are DLS and Zeta potential determined? Which solvent is used for the preparation of the suspension? Clarify
4. Introduction needs to be improved by clearly stating the hypothesis of the study.
5. There are many grammatical and sentence errors in the article, and the language organization needs to be improved.
Need some minor improvements in the syntax
Author Response
Dear reviewer,
we thank you for your attention to our research and valuable recommendations, following which we revised the manuscript. Below we provide answers to your comments, and also attach the manuscript file, where the changes corresponding to the new version are highlighted in red.
- The size and shape of nanoparticles play an important role in biocompatibility. TEM data shown in Figure 2A don't provide any valuable information about the size and shape of nanoparticles which is of poor resolution. Authors need to provide better resolution TEM as well as HR-TEM need to be provided.
: Figure 2A was replaced. - Purity of the nanoparticle is not confirmed. Authors may provide XRD and EDX to confirm the purity of the nanoparticles.
: The formation of the face-centered cubic phase of the metal was confirmed by the electron diffraction pattern according to the ratios of diffraction ring diameters. The ratios of diffraction ring diameters corresponding to {111}, {200}, {220}, {311}, and {222} planes were √3: √4: √8: √11: and √12, respectively. The diffuseness of the diffraction rings was due to the small particle size. The absence of extraneous reflexes indicated the absence of an admixture phase of oxides, hydroxides, or complex metal compounds (see Figure 2A). Diffraction rings are diffuse due to the small particle size. In another way, the efficiency of Pt reduction was evaluated using the reaction in which a colored complex of Pt ions with iodide ions is formed in acidic conditions (see Ref. [38]). XRD will not give advantages over electron diffraction, and 1-3 broadened peaks will also be observed due to the small particle size (less than 5 nm). In another way, the efficiency of Pt reduction was evaluated using the reaction in which a colored complex of Pt ions with iodide ions is formed in acidic conditions [38]. - How are DLS and Zeta potential determined? Which solvent is used for the preparation of the suspension? Clarify
: corresponding text was added to 2.3 section: “A photonic particle analyzer Zetasizer Nano ZS (Malvern, UK) was used to determine the size and zeta potential of the particles. The analyzer has a particle measurement range from 0.6 to 6000 nm. The operating temperature range is 2°C – 120°C, the angle of detection of scattered light is 173°C, a helium-neon laser with a wavelength of 633 nm is used as a light source, the power of the light source is 5 MW. The device determines particle sizes by measuring the rate of fluctuation of scattered light by particles. The measurement was carried out in automatic mode according to the standard procedure. The glass cuvette was filled with 1 ml of the sample, loaded into the cuvette compartment of the device. The beam of light emitted by the laser passes through the attenuator and enters the sample cell. The light scattered by the particles is detected by the detector. The electrical signal of the detector, proportional to the light intensity, is processed by the correlator according to mathematical algorithms embedded in the software. When determining the zeta potential, an immersion-type electrode is lowered into a cuvette filled with a sample. The sample is exposed to an electric field, and the electrophoretic mobility of particles in an electric field is used to calculate the zeta potential. The solvent was water.” - Introduction needs to be improved by clearly stating the hypothesis of the study.
: this section has been revised in accordance with the reviewer's comment. We have added a number of propositions to support our hypothesis. First, these are the basic concepts of therapy for acute disorders of cerebral blood flow: “To date, there are essentially two concepts for the treatment of acute CBF disorders. The first involves the use of thrombolytics (recombinant tissue plasminogen activator) or endovascular thrombectomy, which enables recanalization and restoration of blood flow in damaged vessels. The effectiveness of this approach in practice is significantly limited by the time window available (up to 3-4.5 hours for thrombolysis and 6-24 hours for endovascular thrombectomy), and the possibility of severe complications or contraindications after use [5,6]. The second concept involves the use of various agents that have a neuroprotective effect.
The brain has a high rate of oxygen consumption, and unlike most organs, it has no internal energy reserves. This causes a rapid development of oxidative stress (OS) in the brain, a process in which the generation of reactive oxygen species (ROS) prevails over the ability to eliminate them, both under conditions of ischemia and during subsequent reperfusion. Also, cerebral ischemia causes rapid and strong activation of the sympathetic nervous system [7], which, apparently, triggers the development of systemic endothelial dysfunction (ED) [8]. Due to the key role of OS in the development of ischemic brain damage [9], ED [10], and postischemic hypoperfusion [11], the use of antioxidants is one of the promising areas of neuroprotection. For example, there have already been a few clinical studies showing the effectiveness of the low-molecular-weight antioxidants edaravone [12] and N-acetylcysteine [13].”
Second, we described in more detail the role of aminothiols as markers of cerebral ischemia: “Low-molecular-weight aminothiols (cysteine—Cys, glutathione—GSH, homocysteine—Hcy, and others) are highly sensitive to OS. In response to cerebral ischemia, there is a rapid decrease in the content of their reduced forms in blood plasma, as well as a drop in the redox status of the main intracellular antioxidant, GSH, in the nervous tissue itself [19,20]. GSH has previously been shown to be associated with stroke severity [21]. The important protective role of GSH is also confirmed by the efficacy of using N-acetylcysteine (a readily available substrate for GSH synthesis) in models of ischemia [22,23] and in the treatment of stroke [13]. An elevated Hcy level, which damaging effect is associated with the induction of OS [24], is considered a risk factor for stroke since a decrease in Hcy level can reduce this risk [25]. The effect of nPt on the homeostasis of aminothiols in cells is still poorly understood. Previously, it was shown that nPt decreased the cellular GSH level, and this effect correlated with the particle size in an inverse manner but appeared not to be based on the formation of ROS [26]. To the best of our knowledge, studies on the effect of nPt on aminothiols under stress conditions have not yet been carried out.
Under ischemia–reperfusion injury, the development of OS leads to the activation of a number of transmethylases in the nervous tissue, which manifests as hypermethylation of DNA, non-coding RNA, and histones, and disruption of the biosynthesis of polyamines and acetylcholine, which, in turn, enhance inflammatory damage to neurons [27]. During these transmethylation reactions, S-adenosylmethionine (SAM) is consumed and S-adenosylhomocysteine (SAH) is formed. Thus, the SAM/SAH ratio is referred to as the global methylation index [28]. Numerous works involving models of cerebral ischemia have shown a protective effect of SAM administration [29-31], including decreasing the GSH content in the brain [32]. But studies on the effect of nPt on the balance of SAM and SAH have not yet been conducted.
Thirdly, we also mentioned the problem of bioavailability of neuroprotectors, which will play an important role in the manuscript: “In the study on the neuroprotective effect of substances, one of the key issues is their bioavailability, i.e., the ability to penetrate through the blood–brain barrier (BBB), which is a complex of membranes and intercellular contacts formed by the endothelium of blood vessels, astrocytes, and pericytes [35]. The BBB prevents the transfer of many substances from the blood to the brain. Although it has previously been shown that metal nanoparticles, in particular gold, penetrate the BBB through the mechanisms of passive diffusion, carrier-mediated transport, adsorptive-mediated endocytosis, or pinocytosis [36], there are no data yet on how the BBB can be made permeable to nPt.”
The formulation of the hypothesis has undergone correction: “Thus, based on the neuroprotective effect of nPt in vivo, we set out to shed light on 1) whether nPt is able to effectively suppress systemic OS and postischemic hypoperfusion induced by acute impairment of cerebral blood flow, and 2) whether the effect of nPt is due to a direct effect on brain tissue or due to the extracerebral effect of nanoparticles. To achieve these objectives, we determined indicators of systemic (reduced plasma levels of aminothiols) and brain (reduced GSH, SAM, and SAH in the hippocampus) OS and CBF and brain nPt content using a model of temporary global cerebral ischemia in rats.”
5. There are many grammatical and sentence errors in the article, and the language organization needs to be improved.
Ans.: the manuscript has been subjected to language correction to eliminate errors.

Reviewer 3 Report
Dear Authors,
You did excellent work with your article. Your manuscript offers very insightful information regarding the neuroprotective properties of platinum nanoparticles on a model of global cerebral ischemia in rats. The study's results suggested that platinum nanoparticles did not accumulate in the brain and that their effect on the brain is temporary.
Although, in order to improve the quality of the manuscript, please consider some constructive suggestions:
-The introduction and materials and methods sections should include more recent works (references relevant to the research but after the year 2020).
-Also, some figures could be improved in order to have a more visible image: Figure 2 (B) & (C).
-In the end, the conclusion section could be enriched with more detailed phrases regarding relevant ideas conducted/ formed after the research.
Overall, the study presented in the paper has great potential!
Keep up the great work!
Author Response
Dear reviewer, we thank you for your attention to our research and valuable recommendations, following which we revised the manuscript. Below we provide answers to your comments, and also attach the manuscript file, where the changes corresponding to the new version are highlighted in red.
-The introduction and materials and methods sections should include more recent works (references relevant to the research but after the year 2020).
Ans.: this section has been extensively revised, links to recent studies have been added. First, these are the basic concepts of therapy for acute disorders of cerebral blood flow: “To date, there are essentially two concepts for the treatment of acute CBF disorders. The first involves the use of thrombolytics (recombinant tissue plasminogen activator) or endovascular thrombectomy, which enables recanalization and restoration of blood flow in damaged vessels. The effectiveness of this approach in practice is significantly limited by the time window available (up to 3-4.5 hours for thrombolysis and 6-24 hours for endovascular thrombectomy), and the possibility of severe complications or contraindications after use [5,6]. The second concept involves the use of various agents that have a neuroprotective effect.
The brain has a high rate of oxygen consumption, and unlike most organs, it has no internal energy reserves. This causes a rapid development of oxidative stress (OS) in the brain, a process in which the generation of reactive oxygen species (ROS) prevails over the ability to eliminate them, both under conditions of ischemia and during subsequent reperfusion. Also, cerebral ischemia causes rapid and strong activation of the sympathetic nervous system [7], which, apparently, triggers the development of systemic endothelial dysfunction (ED) [8]. Due to the key role of OS in the development of ischemic brain damage [9], ED [10], and postischemic hypoperfusion [11], the use of antioxidants is one of the promising areas of neuroprotection. For example, there have already been a few clinical studies showing the effectiveness of the low-molecular-weight antioxidants edaravone [12] and N-acetylcysteine [13].”
Second, we described in more detail the role of aminothiols as markers of cerebral ischemia: “Low-molecular-weight aminothiols (cysteine—Cys, glutathione—GSH, homocysteine—Hcy, and others) are highly sensitive to OS. In response to cerebral ischemia, there is a rapid decrease in the content of their reduced forms in blood plasma, as well as a drop in the redox status of the main intracellular antioxidant, GSH, in the nervous tissue itself [19,20]. GSH has previously been shown to be associated with stroke severity [21]. The important protective role of GSH is also confirmed by the efficacy of using N-acetylcysteine (a readily available substrate for GSH synthesis) in models of ischemia [22,23] and in the treatment of stroke [13]. An elevated Hcy level, which damaging effect is associated with the induction of OS [24], is considered a risk factor for stroke since a decrease in Hcy level can reduce this risk [25]. The effect of nPt on the homeostasis of aminothiols in cells is still poorly understood. Previously, it was shown that nPt decreased the cellular GSH level, and this effect correlated with the particle size in an inverse manner but appeared not to be based on the formation of ROS [26]. To the best of our knowledge, studies on the effect of nPt on aminothiols under stress conditions have not yet been carried out.
Under ischemia–reperfusion injury, the development of OS leads to the activation of a number of transmethylases in the nervous tissue, which manifests as hypermethylation of DNA, non-coding RNA, and histones, and disruption of the biosynthesis of polyamines and acetylcholine, which, in turn, enhance inflammatory damage to neurons [27]. During these transmethylation reactions, S-adenosylmethionine (SAM) is consumed and S-adenosylhomocysteine (SAH) is formed. Thus, the SAM/SAH ratio is referred to as the global methylation index [28]. Numerous works involving models of cerebral ischemia have shown a protective effect of SAM administration [29-31], including decreasing the GSH content in the brain [32]. But studies on the effect of nPt on the balance of SAM and SAH have not yet been conducted.
Thirdly, we also mentioned the problem of bioavailability of neuroprotectors, which will play an important role in the manuscript: “In the study on the neuroprotective effect of substances, one of the key issues is their bioavailability, i.e., the ability to penetrate through the blood–brain barrier (BBB), which is a complex of membranes and intercellular contacts formed by the endothelium of blood vessels, astrocytes, and pericytes [35]. The BBB prevents the transfer of many substances from the blood to the brain. Although it has previously been shown that metal nanoparticles, in particular gold, penetrate the BBB through the mechanisms of passive diffusion, carrier-mediated transport, adsorptive-mediated endocytosis, or pinocytosis [36], there are no data yet on how the BBB can be made permeable to nPt.”
The formulation of the hypothesis has undergone correction: “Thus, based on the neuroprotective effect of nPt in vivo, we set out to shed light on 1) whether nPt is able to effectively suppress systemic OS and postischemic hypoperfusion induced by acute impairment of cerebral blood flow, and 2) whether the effect of nPt is due to a direct effect on brain tissue or due to the extracerebral effect of nanoparticles. To achieve these objectives, we determined indicators of systemic (reduced plasma levels of aminothiols) and brain (reduced GSH, SAM, and SAH in the hippocampus) OS and CBF and brain nPt content using a model of temporary global cerebral ischemia in rats.”
-Also, some figures could be improved in order to have a more visible image: Figure 2 (B) & (C).
Ans.: we have updated this figure, trying to improve its quality.
-In the end, the conclusion section could be enriched with more detailed phrases regarding relevant ideas conducted/ formed after the research.
Ans.: we have made some changes to the conclusion as recommended by the reviewer:“ In the present study, we synthesized and purified nPts to demonstrate their effectiveness as neuro- and vasoprotective agents in a tBCAO model in rats. The tBCAO model was not complicated by the presence of products resulting from incomplete Pt reduction. We found that purified nPt prevented the apoptotic death of hippocampal neurons, eliminated the effect of postischemic hypoperfusion, and prevented the decline in the levels of reduced aminothiols in blood plasma. Moreover, nPt attenuated the decline in the GSH redox status and the global methylation index directly in the brain, which indicates the ability of nPt to suppress the development of both local and systemic OS. However, the level of nPt in the brain was significantly lower than that at which the direct antioxidant effects of nanoparticles are observed. This suggests, firstly, that the mechanism of neuroprotective action of nPt is indirect and, secondly, it points to the important role of extracerebral mechanisms in the development of secondary brain damage. Therefore, nPt can be a very useful tool for both therapy and the search for new targets for the treatment of cerebrovascular diseases.”
